Vegetation structure of plantain-based agrosystems determines numerical dominance in community of ground-dwelling ants

Dassou Anicet Gbéblonoudo 1 2 3
Tixier Philippe 3 4
Dépigny Sylvain 2 3
Carval Dominique dominique.carval@cirad.fr 3 5
1 BIORAVE, Faculty of Sciences and Technologies, Dassa, UNSTIM , Benin
2 CARBAP , Douala , Cameroon
3 UPR GECO, CIRAD , Montpellier , France
4 Departemento de Agricultura y Agroforestria, CATIE , Turrialba , Costa Rica
5 UPR GECO, CIRAD , Le Lamentin , France
Minasny Budiman
Electronic publication date: 2017 Nov 13
Publication date: 2017
Volume: 5
Electronic Location ID: e3917
Received 2017 May 23; Accepted 2017 Sep 22
Copyright: ©2017 Dassou et al.
Copyright year: 2017
Copyright holder: Dassou et al.
License: This is an open access article distributed under the terms of the Creative Commons Attribution License, which permits unrestricted use, distribution, reproduction and adaptation in any medium and for any purpose provided that it is properly attributed. For attribution, the original author(s), title, publication source (PeerJ) and either DOI or URL of the article must be cited.
License URL: https://creativecommons.org/licenses/by/4.0/

Keywords: Cameroon, Dominant, Subdominant, Subordinate, Ants, Vegetation strata, Baits

Funding: CIRAD: AIRD grant C2D project E.U. FEDER 30411 This work is part of a PhD thesis of Anicet Dassou and was funded by CIRAD (AIRD grant) and the C2D project. Dominique Carval was funded by the Project ‘sustainable cropping systems design’ from E.U. FEDER (grant PRESAGE no. 30411). Dominique Carval, Philippe Tixier and Sylvain Dépigny were funded by the CIRAD. There was no additional external funding received for this study. The funders had no role in study design, data collection and analysis, decision to publish, or preparation of the manuscript.

==============================
In tropics, ants can represent an important part of animal biomass and are known to be involved in ecosystem services, such as pest regulation. Understanding the mechanisms underlying the structuring of local ant communities is therefore important in agroecology. In the humid tropics of Africa, plantains are cropped in association with many other annual and perennial crops. Such agrosystems differ greatly in vegetation diversity and structure and are well-suited for studying how habitat-related factors affect the ant community. We analysed abundance data for the six numerically dominant ant taxa in 500 subplots located in 20 diversified, plantain-based fields. We found that the density of crops with foliage at intermediate and high canopy strata determined the numerical dominance of species. We found no relationship between the numerical dominance of each ant taxon with the crop diversity. Our results indicate that the manipulation of the densities of crops with leaves in the intermediate and high strata may help maintain the coexistence of ant species by providing different habitat patches. Further research in such agrosystems should be performed to assess if the effect of vegetation structure on ant abundance could result in efficient pest regulation.

Introduction

In tropics, ants are known to potentially represent the major part of animal biomass (Hölldobler & Wilson, 1990). Moreover, in agrosystems, they are known to be involved in pest regulation and other ecosystem services (Perfecto & Vandermeer, 2006; Philpott & Armbrecht, 2006). Understanding the factors affecting the structure of local ant communities is therefore an important issue in agroecology. The structure of the community may be related to physical factors that affect physiology of organisms (humidity and temperature) and ecological factors (Philpott & Armbrecht, 2006). Ecological factors, which are the focus of the present study, can include both ecological interactions (e.g., foraging interference) and habitat-related factors (e.g., nesting sites).

Previous studies have shown that vegetation may affect the ant communities by affecting habitat structure (Perfecto & Vandermeer, 1996; Vasconcelos et al., 2008; House et al., 2012; Murnen, Gonthier & Philpott, 2013). A common observation of these studies is that habitats that reduce the abundance of a dominant ant species increase ant species richness. Perfecto & Vandermeer (1996) showed that the addition of artificial shade to a tropical agrosystem decreased the abundance of the dominant ant Solenopsis geminata while it increased the abundance of other ant species. Vasconcelos et al. (2008) found that trees and tall grasses affect ant species composition in savannas of South America; more specifically, they reported that tall grass cover reduced the incidence of the dominant ant species, Solenopsis substituta. In a study of ants in an agricultural matrix, House et al. (2012) found that species richness and abundance were higher in native woodlands than in pastures or crops but the dominance of Dolichoderinae ants was higher in pastures or crops than in native woodlands. By manipulating food and nesting site availability, Murnen, Gonthier & Philpott (2013) demonstrated that ant community composition is greatly influenced by habitat type, which determines nesting resource availability, while food quantity alone had no effect on community composition.

Ant diet varies within and between subfamilies and genera. Many ants may be mainly omnivorous and opportunistic, while others are specialized for predation, fungus-growing, or herbivory (seeds and nectar) (Hölldobler & Wilson, 1990). Therefore, at the community level, ant diets represent a continuum between herbivory and strict predation (Bluthgen, Gebauer & Fiedler, 2003) and are likely to be affected by plant diversity. Bluthgen, Gebauer & Fiedler (2003) proved through isotope analysis that the dominant ant species with small to intermediate colonies in tree canopies tend to be herbivorous (including feeding on extrafloral and floral nectaries), that the dominant canopy ants with large colonies tend to be omnivorous, and that understorey or ground-dwelling ants tend to occupy higher trophic levels.

In the humid tropics of Africa, plantains (Musa AAB genome) are cropped in association with annual crops (root, tuber, and vegetable crops) and perennial crops (cocoa, coffee, and palm) (Côte et al., 2010). Because such agrosystems differ greatly in vegetation diversity and structure, they are useful for studying how habitat-related factors affect ant community structure. Using diversified plantain agrosystems in the current study, we (i) determined the dominant and subordinate ant species in the dry and rainy seasons and (ii) tested the hypotheses that local vegetation structure and plant diversity determine the numerically dominant ant at the genus level.

Methods

Fields, plots, and subplots

We conducted our study in the Moungo department of the Littoral Region of Cameroon (Central Africa) from June 2012 to February 2013. We selected 20 farmer fields near the CARBAP research station (4°34′11.33″N; 9°38′48.96″E; 79 m a.s.l.). Field experiments were approved by the CARBAP Research Station (Njombé, Cameroon) where the experiment was performed. All the fields have a young, brown soil derived from a volcanic platform (Delvaux, Herbillon & Vielvoye, 1989). The climate is humid tropical with a monthly mean temperature ranging from 25.0 to 27.4 °C and a mean annual rainfall of 2,610 mm. All fields contained plantain crops (Musa AAB genome) and a diverse array of other annual and perennial crops. Pesticides and fertilizers are rarely applied in these low input agrosystems.

In each field, we assessed ants and crops in one 12 × 12 m plot, which was subdivided into 25 subplots of 2.4 × 2.4 m. We sampled during two periods: the rainy season (mid-March 2012 to mid-November 2012) and the dry season (mid-November 2012 to February 2013).

Vegetation structure and diversity

For all subplots, we identified all cropped plants, measured their density (number of plants of each species per m2), and recorded their coordinates with a measuring tape (using subplot corners as a references to minimize error). We classified the plant species into four categories according to the location of their canopies relative to the soil surface: low stratum (height ≤ 2 m), intermediate stratum (2 m < height ≤ 6 m), high stratum (height > 6 m), and Musa group. For each category, we calculated the density of plants, i.e., the number of plants of a considered category per m2. Plant diversity at each subplot was assessed by the Shannon Index (Shannon, 1948), which was calculated with the ‘diversity’ function of the ‘VEGAN’ R package (Dixon, 2003).

Bait sampling

In each subplot, we measured ant abundance by using 2/3 tuna–1/3 honey baits. The 2 cm-radius bait was placed in the centre of a white ceramic square tile (30 cm side), which was itself placed at the ground level in the centre of the subplot. Thirty minutes after the baits were deployed, we counted the individuals of different species/morphospecies present on the tile. Samples of all observed species were collected and conserved in 70% alcohol to perform identification to the genus according to Fisher & Bolton (2016), then to the species. When we were not able to determine the species, a morphospecies number was assigned to the individual on the basis of morphological specificities. The ants were also recorded according to a six point abundance scale (following Andersen, 1997; Parr et al., 2005; Baccaro, Ketelhut & De Morais, 2010). We performed bait samples twice for each subplot, during two periods: the rainy season (mid-March 2012 to mid-November 2012) and the dry season (mid-November 2012 to February 2013).

Dominant, subdominant and subordinate ants

Following Baccaro, Ketelhut & De Morais (2010) and similarly to Carval et al. (2016), we combined three numerical and behavioral criteria of dominance to determine dominant, subdominant and subordinate ants. The dominant (respectively subdominant) ants were considered as those that were recorded in >10% of all baits, controlled >25% (respectively >10%) of baits where they occurred, and with a mean abundance score (i.e., the sum of the abundance scores for the species at all baits divided by the number of baits at which the species was present) of >3.5 (respectively >3). All other species that did not meet all these criteria was considered as subordinate species.

Then, we grouped ants by genus and we excluded Odontomachus troglodytes from the following analysis because of its very low occurrence on baits (see Table 1). We assessed the influence of the season (dry, rainy) on the occurrence of each genus by using binomial generalized linear models.

Table 1 Occurrence of dominant, subdominant, and subordinate ants at baits.

Species	Baits recorded (%)	Baits controlled (%)	Mean abundance score	
	Rainy season	Dry season	Rainy season	Dry season	Rainy season	Dry season	
Dominant	
Pheidole spp.	36.8	43.6	25.6	37.2	3.4	3.8	
Subdominant	
Axinidris murielae	0.0	10.4	–	13.5	–	4.2	
Subordinate	
Paratrechina longicornis	32.6	36.8	11.0	9.2	2.8	2.8	
Tetramorium sp.	11.2	13.4	7.1	20.1	2.4	3.2	
Monomorium bicolor	16.0	7.8	12.5	15.4	2.7	2.5	
Monomorium sp. 1	25.6	28.2	4.7	3.5	2.4	2.1	
Monomorium sp. 2	0.0	1.2	–	0.0	–	2.5	
Camponotus acvapimensis	30.2	29.8	1.3	5.4	2.3	2.2	
Camponotus brutus	22.2	15.2	9.9	7.9	2.5	2.3	
Camponotus sp. 1	0.0	1.2	–	33.3	–	3.3	
Camponotus sp. 2	0.0	0.6	–	0.0	–	3.7	
Odontomachus troglodytes	6.2	5.6	0.0	3.6	1.5	1.7	

Effect of vegetation strata on numerical dominance of ants

For each subplot, we attributed rank values for each ant genus according to their respective abundances (Parr & Gibb, 2010). The genera with the rank of one were considered as the numerically dominant genus at the subplot scale. Then, we used multinomial logit model to assess the effect of plant diversity and of the density of each stratum on the probability that an ant genus was numerically dominant. We used likelihood ratio tests (LRTs) to select the best model by removing non-significant parameters in a backwards-stepwise process. The selection procedure was continued until a model was found in which all effects were significant (Zuur et al., 2009). Multinomial models were estimated using the ‘VGAM’ package (Yee, 2010).

All statistical analyses were performed with R 3.3.1 (R Core Team, 2016) and with an alpha level of 0.05.

Results

Overall, we recorded 20,910 ants belonging to 11 species or morphospecies. Pheidole spp. was the most abundant taxon (9,200 individuals) followed by Paratrechina longicornis (3,037 individuals), Monomorium sp. 1 (1,696 individuals), Tetramorium sp. (1,562 individuals), Camponotus acvapimensis (1,517 individuals), Camponotus brutus (1,328 individuals), Monomorium bicolor (1,296 individuals) and Axinidris murielae (895 individuals). The remaining four species were relatively scarce, namely: Camponotus sp. 1 (166 individuals), Odontomachus troglodytes (144 individuals), Monomorium sp. 2 (35 individuals) and Camponotus sp. 2 (34 individuals).

Dominant, subdominant, and subordinate ants

Pheidole spp. was identified as the dominant genus because it combined a high occurrence on baits, a large proportion of controlled baits and a high mean score abundance (Table 1). Axinidris murielae was identified as a subdominant species because it combined a moderate proportion of controlled baits and a high mean score abundance (Table 1). All other species were considered subordinate (Table 1).

Occurence of each genus was not significantly affected by the season, except for Axinidris murielae which was absent on baits in the rainy season and for Pheidole spp. whose occurrence was higher in the rainy season (Figs. S1 & S2, Table S1). Frequency of numerical dominance was similar in the rainy season and dry season (Fig. 1).

Figure 1 Frequencies of numerical dominance of subplots for each ant taxon in the (A) rainy and (B) dry seasons.

Effect of vegetation strata on numerical dominance of ants

We recorded 31 plant species, which we grouped into four vegetation strata (Table 2). The probability of dominance of each ant taxa was not significantly affected by the density of plants in the low and Musa strata but was significantly affected by the density of plants in the intermediate and high strata (Table 3). The dominance of Pheidole spp., Monomorium spp., and Tetramorium sp. was negatively correlated with the density of plants in the intermediate and high strata, whereas the dominance of P. longicornis, Camponotus spp., and A. murielae was positively correlated with the density of plants in the intermediate and high strata (Fig. 2). The probability of dominance of each ant taxa was not significantly correlated with plant diversity (Table 2).

Table 2 Cultivated plant species in each stratum of diversified plantain-based agroecosystems.

Stratum refers to the location of the plant canopy relative to the soil surface.

Stratum	Cultivated plant species	
Low	Arachis hypogaea L. (groundnut), Xanthosoma sagittifolium (Schott) (macabo), Colocasia esculenta L. (taro), Dioscorea spp. (yam), Capsicum anuum L. (hot pepper), Solanum macrocarpon L. (garden egg), Corchorus spp. (crin-crin), Ananas comosus L. (pineapple), Amaranthus spp., Solanum lycopersicum L. (tomato), Abelmoschus esculentus (Medik) (gombo), Vigna unguiculata L. (cowpea), Ipomoea batatas L. (sweet potato), Zea mays L. (maize)	
Intermediate	Carica papaya L. (papaya), Manihotesculenta (Crantz) (cassava), Vernonia spp., Gnetum africanum (eru), Triumphetta pentadra (Rich.)	
High	Elais guineensis (Jacq.) (oil palm), Coffea Arabica L. (coffee), Theobroma cacao L. (cocoa), Cola acuminata (Schotte & Endl.) (cola), Dacryodes edulis Lam (safou), Persea americana (Mill.) (avocado), Psidium guajava L. (guava), Mangifera indica L. (mango)	
Musa	Musa AAA (banana), Musa AAB (plantain)	

Table 3 Likelihood ratio tests for the strata multinomial model.

Stratum refers to the location of the plant canopy relative to the soil surface. Intermediate, high, and low strata indicate a high density of plants with canopies at intermediate, high, and low strata, respectively. Values in bold are statistically significant at an alpha level of 0.05.

Variable	Δd.f.	Chi2	p-value	
Intercepts	5	333.29	<0.0001	
Plant diversity	5	7.68	0.174	
Intermediate stratum	5	33.14	<0.0001	
High stratum	5	18.85	0.002	
Musa stratum	5	10.00	0.075	
Low stratum	5	9.96	0.076	

Figure 2 Predicted probability of dominance for (A) P. longicornis, (B) Pheidole spp., (C) Tetramorium sp., (D) A. murielae, (E) Camponotus spp. and (F) Monomorium spp.

Grey curves: response to plant density of intermediate stratum; black curves: response to plant density of high stratum.

Discussion

We found that ants of the Pheidole genus were the numerically dominant ants in our study fields. Abera-Kalibata et al. (2007) found that three morphospecies of Pheidole were among the most abundant ants in banana fields in Uganda. Elsewhere, we observed similar frequencies of numerical dominance for P. longicornis, Camponotus spp., and Monomorium spp. These results also agree with the literature in that ants of the Camponotus genus are considered ubiquitous subordinate ants that may numerically dominate arboreal vegetation (Davidson, 1997; Tadu et al., 2014). The tramp crazy ant P. longicornis is an exploitative competitor and uses a foraging strategy with worker recruitment occurring at a short-range of distance (Kenne et al., 2005). The numerical dominance of P. longicornis on baits is thought to be principally linked to its foraging speed (Kenne et al., 2005). Tetramorium sp. and A. murielae were numerically dominant less frequently than the other taxa. However, when present on baits, A. murielae displayed a high abundance score resulting in the control of a moderate proportion of baits.

We hypothesized that the vegetation structure determines which species numerically dominates the ground-dwelling ant community at the local (subplot) scale. We indeed found that the general trend of numerical dominance can be altered by the density of plants in the intermediate and high strata. The probability of being numerically dominant for ground-dwelling ants like Pheidole spp., Monomorium spp., and Tetramorium sp. decreased as the density of the intermediate and high strata increased, while the probability of being dominant for the mostly arboreal taxon Camponotus spp. and the tramp species P. longicornis increased with the density of plants in the intermediate stratum. A high density of high strata plants also increased the abundance of these taxa, but as the density of plants with leaves in the high stratum increased, the dominance of the strictly arboreal ant A. murielae increased. We found no effect of plant density in low stratum on the dominance of ants. In Australia, Stevens et al. (2007) also found no effect of ground cover on the dominance of the Dolichoderinae ant Iridomyrmex in citrus groves. Together, these results suggest that plant density in the low stratum does not directly modify habitats for the six studied taxa (Andersen, 1995). However, the low stratum may have influenced the cryptic ants (e.g., hypogaeic and litter-dwelling ants), as demonstrated by Bestelmeyer & Wiens (1996); that possibility should be investigated in future research.

According to Ribas et al. (2003), low and high woody plant densities may influence ant communities through three processes: (i) resources increase with woody plant density, and an increase in resources would enhance ant species diversity; (ii) habitat conditions are altered by the density of woody plants, and habitat conditions would affect which ants are numerically dominant; and (iii) the variation in woody plant densities may lead to species–area patterns. Our results on dominance hierarchies are in agreement with the second and third processes. Indeed, the effects of strata densities are consistent with the preferred ecological niches of the six studied ant taxa. For instance, ground-dwelling taxa were, in our study, negatively related to the density of arboreal habitats (e.g., intermediate and high strata). This agrees with Lassau & Hochuli (2004) who found that the abundance of species that only nest on ground was negatively related to the density of tree cover. The abundance of Camponotus spp., which may forage both on the ground and in the arboreal stratum, was positively related to the density of high strata plants, which correspond to arboreal nesting or foraging habitats, except in the extreme densities of the high stratum, which coincided with the numerical dominance of A. murielae The members of the latter species nest strictly in trees and are primarily arboreal foragers but may occasionally forage in ground litter (Snelling, 2007). We observed individuals of A. murielae on baits only in the dry season, which is consistent with the view that arboreal ant species forage at ground-level during the dry season, when resources in trees are relatively scarce (Delabie, Agosti & Do Nascimento, 2000). P. longicornis, known as the crazy ant, is a native of West Africa and prefers moist habitats for reproduction (Kenne et al., 2005). The nests of this tramp species are often small, ephemeral and occur in a wide range of habitats (e.g., plant cavities, live or dead plants, leaf litter). An increase in the density of plants with leaves in the intermediate and high strata may enhance the local hygrometry and therefore increase the nesting sites available for P. longicornis. However, P. longicornis is a weak competitor against common ground-dwelling ant species (including Camponotus spp.) in its native range (Kenne et al., 2005). We hypothesize that, as the density of plants with leaves in the high stratum increases in a plantain field, the availability of foraging and nesting sites increases, and better competitors like Camponotus spp. and Axinidris murielae predominate the area and resulting in a decrease in the positive effect of the intermediate stratum density on P. longicornis.

We found no relationship between the numerical dominance of each ant taxon with the crop diversity. One explanation may be that the studied taxon were omnivores that feed in multiple trophic level (consumers of plant resources, hemipteran honeydew, herbivores, predatory arthropods or even scavengers), and may not be affected by the identity of plants that support only a part of their diet. One other explanation may be that the presence and abundance of species is linked to nesting habits. For instance, most Camponotus spp. forage both arboreally and on the ground but have specialized nesting habits in that they generally start colonies in living or dead trunks, such as banana pseudostems. Davidson (1997) argued that this kind of ant species locates its nest on preferred resource plants. Consequently, plant diversity would not modify their nesting or foraging habits.

Ants have been increasingly recognized as important predators in tropical and subtropical agricultural systems (Way & Khoo, 1992; Perfecto & Castineiras, 1998; Offenberg, 2015). Ants have complex and often strong effects on lower trophic levels (Philpott, Perfecto & Vandermeer, 2008) and may be useful in pest management (Perfecto, 1991). In plantain and banana agrosystems, the banana weevil Cosmopolites sordidus (Germar) (Coleoptera: Curculionidae) is the most important pest (Gold, Pena & Ekaramura, 2001). In Martinique, using metabarcoding analysis and predation tests, Mollot et al. (2014) recently showed that C. sordidus is preyed on by the arboreal ant Camponotus sexguttatus F. (Hymenoptera: Formicidae) and the ground-dwelling ant Solenopisis geminata. In the current study, we have shown that Camponotus spp. were favoured by the intermediate and high strata. Pheidole spp. has been suggested to be a potential natural enemy of C. sordidus in Uganda (Abera-Kalibata et al., 2007; Abera-Kalibata, Gold & Van Driesche, 2008), and Pheidole megacephala and Tetramorium guineense (Bernard) (Hymenoptera: Formicidae) are used as biological control agents of C. sordidus in Cuba (Castineiras & Ponce, 1991; Perfecto & Castineiras, 1998). Our results indicate that the manipulation of the densities of crops with leaves in the intermediate and high strata may help maintain the coexistence of ant species by providing different habitat patches. Further research in such agrosystems should be performed to assess if the effect of vegetation structure on ant abundance could result in efficient pest regulation.

Supplemental Information

Data S1 Raw data

Click here for additional data file.

Supplemental Information 1 Multinomial analysis R-script

Click here for additional data file.

Supplemental Information 2 Supplemental figures and table

Click here for additional data file.

We thank B Jaffee for revising English language of the manuscript.

Additional Information and Declarations

Competing Interests

Author Contributions

Field Study Permissions

Data Availability

The authors declare there are no competing interests.

Anicet Gbéblonoudo Dassou conceived and designed the experiments, performed the experiments, analyzed the data, wrote the paper, prepared figures and/or tables, reviewed drafts of the paper.

Philippe Tixier conceived and designed the experiments, analyzed the data, wrote the paper, reviewed drafts of the paper.

Sylvain Dépigny conceived and designed the experiments, performed the experiments, wrote the paper, reviewed drafts of the paper.

Dominique Carval analyzed the data, wrote the paper, prepared figures and/or tables, reviewed drafts of the paper.

The following information was supplied relating to field study approvals (i.e., approving body and any reference numbers):

Field experiments were approved by the CARBAP Research Station (Njombe, Cameroon) where the experiment was performed.

The following information was supplied regarding data availability:

The complete raw data with information on vegetation (density of each crop species) and on each ant species (abundance) in contained in a Supplemental File. A commented R script that enables the reproduction of all the statistical analysis and all the figures is also uploaded as a Supplemental File.

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
