# Peer review of "Vegetation structure of plantain-based agrosystems determines numerical dominance in community of ground-dwelling ants"

_PeerJ, doi:10.7717/peerj.3917_

## Round 0.1 · original submission · Major Revisions

· Academic Editor

Major Revisions

This is an interesting study, however 2 of the reviewers are particularly concerned about the lack of taxonomical detail for the ant species. This needs to be fully addressed. The statistical method on response curve (Fig 2) needs further discussion.

·

Basic reporting

This manuscript is generally well written and the data have been subjected to sound statistical analysis. There are some textual flaws and I have annotated these on the manuscript.
I am not entirely comfortable with the use of the term 'plantain based agroecosystem'. If cocoa or coffee is present then aren't such farms likely to be cocoa or coffee farms with some plantain interplanted? In other words, is plantain always the main crop in the farms that were studied?
What concerns me is the lack of ant identifications to species level, with one exception. I suspect that the Phedole app. are members of the megacephala group. As presented, the reader is unsure whether this study represents a set of native ants or whether the fields are colonised by 'weedy', cosmopolitan ant species. I would have thought that this was essential information.

Experimental design

Since the ultimate thrust of this work is to see if farms harbour useful biological control ant agents, this study has missed an important opportunity to include the sampling of arboreal ants. Baits could easily have been placed in the plantain, although I doubt if plants would occur in each of the 25 subplots. This would therefore demand a somewhat different analysis.
As this manuscript stands, the are copious implications about arboreal ants and control of tree pests, without actually having looked at the arboreal system. Maybe this has been done in the thesis, but if so surely some data could be brought into this manuscript.

Validity of the findings

The findings presented here are valid but not as profound as they could be due to a) lack of species-level identifications and b) lack of data on arboreal ants.

Additional comments

lines 129-133. It is not clear how you qualified dominant versus subdominant species. I realise that this is embedded in the text here, but you really need a separate sentence for definition of dominant species and another for subdominant species.
Figure 1. This should be in colour as the grey tones are not well differentiated.
Line 201. Refers to functional groups, yet nothing is said about this in the Methods nor in the results.
Lines 231-232 and elsewhere in the Discussion. You refer to species dominating the area, yet you have previously dismissed most of these species as being dominants. Change terminology to ''predominating in the area' or something else that does not clash with the word dominant.
Many other editorial matters are marked up on the PDF.

Reviewer 2 ·

Basic reporting

This is a manuscript about how vegetation densities at high and intermediate strata can change the numerically dominant ground foraging ant at a local scale in plantain polycultures. For the most part, this manuscript uses clear, unambiguous, and professional language. The Figures are relevant and layout is fine. Raw data is included.

However, the manuscript could be improved by considering the following:
Line 27: instead for “cropped in association” you may want to consider using the terms “intercropping” or “grown in a polyculture” (also applicable for line 78)
Lines 47-49: It would be clearer if this were rephrased to “Ecological factors, which are the focus of the present study, can include both ecological interactions (e.g. foraging interference) and habitat-related factors (e.g. nesting sites).
In general it would improve readability and make the manuscript more succinct if the study examples were restructured to include the citation at the end. For example Lines 57-60: This sentence would be clearer if rephrased to something like “For example, in South American savannas tall grass cover has been shown to reduce the incidence of the dominant ant species, Solenopsis substituta (Santschi) (Vasconcelos et al. 2008).
Line 73: change “proved” to “showed” or “provided evidence”
Line 76: change “understorey” to “understory”
Line 122: Add a period after “dominance”
Line 127: Remove “of”
Line 133: change “genus” to “taxa”
Line 134: remove “species”
Line 135: change “genus” to “taxa”
Line 154: change “(Table 1)..” to “(Table 1).”
Line 155: change “genus” to “taxa”
Line 158: change “species or genus” to “taxa” and remove “as”
Line 177: should be “Abera-Kalibata et al. (2007) found that two morphospecies of Pheidole were among the most abundant”
Line 183: change “with a recruitments” to “with worker recruitment”
Line 219: change “genera” to “genus”
Line 239, 240, and 243: change “habits” to “strategies”

Experimental design

The manuscript provides original primary research within the aims and scope of the journal, however there are several areas where the methodology needs clarification:

Lines 117-120: Only half of the ant taxa you report are identified to morphospecies in your results, the rest are identified to genus. Your methods should be fixed to reflect this, and explain your reasoning (why not identify them all to morphospecies?).
Line 124: Your abstract (Line 29), indicates that you deliberately excluded less abundant ant species (which would explain what appears to be surprisingly low diversity in your reported ant taxa). Please clarify if that is the case and include it in the methods if it is.
Lines 127-134: Baccaro et al. (2010) used somewhat different dominance parameters than those used in this ms ( >5% bait presence, >25% bait control, and a minimum mean abundance score of 4 compared to your >10% bait presence, >25% bait control, and a minimum mean abundance score of 3.5). Thus with the methods in Baccaro et al. (2010) none of your ant taxa would be “dominant” (Pheidole spp. has a mean abundance of 3.76). However, if you had lowered the bait control parameter instead of the mean abundance score parameter, the Axinidris sp. would have be “dominant”. I think it would improve the manuscript if you provide an explanation for your reasoning in changing these parameters from the original methods and why this measure of dominance is meaningful in general.
Line 121: What do you consider “control” of a bait to be for this study?

Validity of the findings

Validity of the findings:
Many of the findings appear sound, however I have some concerns about the following:
(1) Pheidole is a large genus, and it’s possible that you have a large number of rare species masking as a single abundant taxa, I think this is important to address.
(2) Ant biocontrol of pests occurs at the level of the plant leaves, rather than on the ground, and ant dominance will often be quite different on plants than on the ground, this is also important to address in your discussion section.
(3) “Dominance” has so many meanings, several of which you use throughout your manuscript. I think it would be useful to clarify why you chose to use your two different measures of dominance (rather than just numerical dominance at the local scale and overall abundance at the broadest scale) or conversely why you didn’t consider bait monopolization at the local scale in your vegetation strata analysis.

Additional comments

All general comments are above

Reviewer 3 ·

Basic reporting

Gbéblonoudo et al. have accomplished one manuscripts deals on an interesting topic, with a great effort to collect vegetal information in different strata in plantain crops, however I have some concerns about ant datasets, analyses and description of the results.

I think that english must to be check.

Experimental design

The design is ok but I'm not sure if evaluate dominance of ant species using only genera is the most correct way.i'ts

Validity of the findings

Important results can be found but i think that a the ant taxonomy to specie level shoul be revised.

Additional comments

Gbéblonoudo et al. have accomplished one manuscripts deals on an interesting topic, with a great effort to collect vegetal information in different strata in plantain crops, however I have some concerns about ant datasets, analyses and description of the results.

Line 39: All your Key Words are present in the tittle. I would suggest that you can make better use of this part to give more ideas of your work and results.
Par exemple: Cameroon, Formicidae, density, abundance, baits, vegetation strata, Dominant, subdominant, and subordinate ants, Musa etc, etc……

Introduction
Line 47: humidity and temperature are environmental factors not physiological. Maybe you can say that these factors affects ant physiology…. please rewrite this idea to be consistent.

Line 47: To be more direct this phrase can be re write as ….. “and ecological factors (Philpott and Armbrecht, 2006), that include ecological interactions (e.g., foraging interference) and habitat-related factors (e.g. nesting sites), which are the focus of the present study.
And please if you can, include some references for each factor mentioned.
Line 56: (hymenoptera: Formicidae) can be deleted
Line 60: Idem
Line 62: The phrase: “but that dominance by ants in the Dolichoderinae was higher in pastures” have not sense….. maybe you want to say that: ‘…..but dominance of Dolichoderinae ants was higher in pastures”….

Line 79: Please include a reference at the end of the phrase.

M&M
Line 67: I’m pretty sure that the phrase is ‘ant diet varies….’ instead ‘Ant diets vary…’ Please check the English.
Line 116: Here you speak about species and morphospecies but you don’t give information of them.
Line 119: delete “the”
Line 122: Please you should include one dot before “We”
Line 118: Rewrite as follow: according to Fisher and Bolton (2016)…..
Line 127 to 129: you say that you follow Baccaro et al. 2010 and Carval et al. 2016, but they use the specie and morphospecies level in theirs analysis. Why do you decide to use genera level? Can you explain your decision of use genera instead of specie level?
As far as I know, is important to take in account the variation inside the genera because one genera can have many species with different diets, behavior, etc, for example inside the genera Pheidole, different species can have marked differences in terms of resource exploitation, (one sp. can be more dominant than others) and in my opinion this is an important factor to take in mind.
Also, in some place you should state how many baits did you put on the field, or say the exact number of subplots and say that in each one your put a bait, it’s seem simple but this information is really important.
Results

You must to begin with a general phrase saying the total abundance of species (genera in your case). Also the number maximum the genera par bait or the minumun… etc… this information is also relevant.
Line 154: Please delete one of two dots between (Table 1) and Pheidole…..

Line 154 and 155: ‘Pheidole spp.were identified as the dominant genus’…. Maybe this could be due to the fact that is the most diverse genus?

Line 153 to 158: You should give the data information of how much each one controlled the bait. And know that this in the table 1 too but you can also give data about the number of baits or the number of workers in each bait, etc….

Line 160: “frecuency” instead “frecuencies” please check the English.

Line 164: I’m not sure that the word collected is the most appropriate maybe is ‘recorded’…

Discussion

Line 178: please use a contrast connector between ‘Uganda’ and ‘We’ …. Juste to differentiate two ideas.

Line 184: Please use “on baits” instead “at baits”

Line 191 to 197: And how you explain those results… less/more resources to nest, or to feed, etc????

Line 213: “…..in our study, negatively related to the density of arboreal habitats”… I think that this is due to the sampling method used, this is expected when you use soil baits, but if you use arboreal baits? Or if you follow your soil baits in the time…. In those cases the results can show that different species can visit the bait…. Its important to make the difference….

Line 232: Axinidris sp. dominate the area (Vasconcelos et al., 2008)… I’m not sure that this work can say something about Axinidris (an African ant), when they mainly works in Amazonian landscapes.

Line 235 to 236: See how is important the specie level? You cannot make a generalization in term of diets or trophic level with certain ant genera, as Pheidole.

I like so much the figure 2, its very informative but I think that you show explain it better in the text.

A general conclusion is missing.


Please check also these references, that can be useful:

- Achury et al 2012. Psyque. “Effects of the Heterogeneity of the Landscape and the Abundance of……….”
- LeBrun et al 2007. Ecology. An experimental study of competition between fire ants and argentine ants in their native range

---

## Round 0.2 · Minor Revisions

· Academic Editor

Minor Revisions

The reviewer has suggested further improvements and edits.

·

Basic reporting

The authors have attempted to address the comments of the reviewers. It is now a serviceable piece of work, although a few edits still need to be attended to, including one uncited reference. I have made comments directly on the manuscript, which appear in blue.

I am still not overly excited about the study for the reasons I already mentioned, namely the lack of sampling of vegetation and the lumping of species. This latter has, to an extent, been addressed, but the analysis and figures still treats the species within a genus as lumped together.

Experimental design

Adequate

Validity of the findings

Adequate

Additional comments

The authors have attempted to address the comments of the reviewers. It is now a serviceable piece of work, although a few edits still need to be attended to, including one uncited reference. I have made comments directly on the manuscript, which appear in blue.

I am still not overly excited about the study for the reasons I already mentioned, namely the lack of sampling of vegetation and the lumping of species. This latter has, to an extent, been addressed, but the analysis and figures still treats the species within a genus as lumped together.

---

## Round 0.3 · accepted · Accept

· Academic Editor

Accept

Hope your study will inspire new work on the tropical system